# Monitoring the Effectiveness of a Preschool Hand Hygiene Intervention Using Flow Cytometry to Determine the Reduction in the Microbiological Counts

**DOI:** 10.3390/ijerph22060846

**Published:** 2025-05-28

**Authors:** Samantha Lange, Tobias George Barnard, Nisha Naicker, Atheesha Singh

**Affiliations:** 1Water and Health Research Centre, Faculty of Health Sciences, University of Johannesburg, P.O. Box 17011, Doornfontein, Johannesburg 2028, South Africa; tgbarnard@uj.ac.za (T.G.B.); asingh@uj.ac.za (A.S.); 2Environmental Health Department, Faculty of Health Sciences, University of Johannesburg, P.O. Box 17011, Doornfontein, Johannesburg 2028, South Africa; nishan@nioh.ac.za; 3Epidemiology and Surveillance, National Institute for Occupational Health, National Health Laboratory Services, Braamfontein 2001, South Africa

**Keywords:** hand hygiene, flow cytometry, preschool children, turbidity, diarrhea, handwashing

## Abstract

Hand hygiene (HH)-related illnesses, such as diarrhea, are one of the leading causes of death in children under five, whereas handwashing with soap can reduce infection rates in this age group. This study monitored whether a simple intervention in the form of a handwashing exercise could reduce the pathogens on preschool children’s hands, potentially reducing HH-related diseases. Hand bag-wash samples were collected from preschool children (N = 160) participating in an intervention study. The samples were collected pre- and post-intervention from the intervention group (IG) and control group (CG). The samples were analyzed using flow cytometry, where the microbiological counts and turbidity were compared between the left and right hands, genders, and pre- and post-intervention groups. The results indicated no significant difference in the microbiological counts of the left and right hands or between the genders of participants, with a significant reduction in intact live cells on the IG children’s hands post-intervention (*p* = 0.000). There was a significant positive correlation (*p* = 0.000) between the turbidity pre- and post-handwashing, with a decreased mean in terms of the turbidity recorded for both groups. Handwashing either with or without soap reduced the microbiological counts on preschoolers’ hands. Reinforcing handwashing at critical times and promoting correct handwashing procedures can assist in reducing hand-hygiene-related diseases in preschool children.

## 1. Introduction

Two leading causes of death in children under the age of 5 years in South Africa remain gastric and respiratory infections [1]. Although there are large bodies of evidence to suggest that handwashing with soap can reduce the infection rates of such illnesses, they are still a frequent occurrence and the burdens of these diseases are felt within the child population [2]. Contaminated hands play a major role in pathogenic transmission and it has been established that hands can carry different types of pathogenic material [3]. Several observational studies have shown that children place objects or fingers in their mouths regularly, and hand samples have revealed both animal and human fecal matter on children’s hands, increasing their risk of diarrheal diseases [4,5,6]. Studies have shown that there may be an increased bacterial load on the dominant hand as it is thought that this hand is more likely to be used to pick up objects [7]. According to reports, higher standards of hygiene are normally found in girls under five years of age [8].

Handwashing interventions can reduce diarrheal diseases by as much as 30% [9]. Studies indicate that handwashing with soap compliance (before the COVID-19 pandemic) was as low as 19% worldwide [10] and that children’s hand hygiene (HH) compliance was around 42% [11]. It can be expected that this figure will have improved during 2020–22 as there is research to suggest that fear during pandemics increases compliance with HH practices [12]. Recent studies have shown that compliance wanes over time regardless of external factors, such as pandemics [13,14]. Hand hygiene interventions are well documented and range from providing water, sanitation and hygiene [15] to providing education on hand hygiene with practical demonstrations of how to wash hands [16,17,18]. Proper hand hygiene could prevent hygiene-related diseases in preschool children, and therefore, ensuring proper and effective handwashing amongst preschoolers could assist in preventing or reducing the occurrence of these diseases.

Studies of hand hygiene in low- to middle-income (LMI) countries have stated that people in these countries are 69% more likely to have hands contaminated with *Escherichia coli* than their higher-income-country counterparts [19]. Although South Africa falls within the LMI bracket, StasSA (2024) has reported that 82.7% of South Africans have access to piped water [20]. In the context of this research setting within suburban preschools, where all households have access to piped water, it can be assumed that a far lesser percentage of children may have hands contaminated with *E. coli*, which is similar to the findings in HI countries, where only 6% of people’s hands were contaminated [19]. The unique methodology of data analysis and reporting of hand “bag-wash” sample results in this study explores the possibility that although traditional fecal indicators such as *E. coli* were not found on the children’s hands, the indicators used would show that their hands were either correctly washed to remove bacterial cells or not.

This study investigated whether a simple hand hygiene intervention administered to preschool children aged 4–5 years could reduce the microbiological counts on their hands by analyzing the results of bag-wash samples to determine if the reduction was due to the removal, inactivation, or death of bacteria. Research has shown that bacteria can enter a viable but non-culturable state [21], which could result in overestimating the effectiveness of procedures like handwashing; therefore, flow cytometry was used instead of typical culture-based methods to study the microbial population of the hand wash samples collected with the bag-wash method [22].

The objectives were to compare the microbial counts of the left and right hands, the differences between the microbial counts on boys’ and girls’ hands and the differences in microbial counts between the intervention group (IG), who were administered the hand hygiene intervention, and the control group (CG), who were not.

## 2. Materials and Methods

### 2.1. Study Design

This study was a pre- and post-test conducted at preschools located in the Kempton Park suburbs of the city of Ekurhuleni, South Africa [23]. Considering the resources available, a random sampling approach was used and targeted children attending preschool between the ages of 4 and 5 years. The legal parent, guardian and relevant authority were provided with information pertaining to this study and provided their written informed consent to allow their children to participate in this study. Children gave informed assent by making use of a pictorial representation of the sample process and affixing a sticker to the assent form if they agreed to participate. The study design was approved by the University of Johannesburg’s Faculty of Health Sciences Research Ethics Committee (REC-01-165-2017) and the Ministerial Consent for Non-Therapeutic Research on Minors for Department of Health National Ethics Research Council (GP-201804-002) and was conducted in compliance with procedures approved under a published study protocol [24].

### 2.2. Sample Collection

A controlled intervention study was conducted at randomly selected preschools in Ekurhuleni, Gauteng (N = 17), from February to December 2019. The schools were randomly placed into the intervention (n = 9) and control (n = 10) groups, with 160 children (N = 160) participating pre-intervention (IG n = 81 and CG n = 79) and 148 children (N = 148) participating post-intervention (IG n = 77 and CG n = 71). Bag-wash samples were taken at both the IG and CG schools, with the sample collectors attempting to sample at a similar time in the morning both pre- and post-intervention, while the children were still in the classroom and had not yet moved to the outside play area. In the pre-intervention study, the right- and left-hand bag-wash samples were collected from the participating children separately in sterile polyethylene Ziploc bags containing 70 mL sterile phosphate-buffered saline solution (Sigma Life Sciences, St. Louis, MO, USA; pH 7.02). The children’s hands were sampled during their current classroom activity and each child was required to rinse each hand in the bag for approximately 20 s. A note was made on the gender of each child, according to their sex assigned at birth, as the sample was collected.

The post-intervention sampling occurred two months after the pre-intervention samples were collected and after the intervention was applied to the intervention group. Bag-wash samples were collected from the participating children during their current classroom activity, with the children being required to wash both their hands, by interlocking their digits for approximately 20 s, in a single sterile polyethylene Ziploc bag containing 100 mL sterile phosphate-buffered saline solution (Sigma Life Sciences; pH 7.02). These samples were labeled as “before” samples. The children were then instructed to wash their hands as they normally would and return for an “after” sample. The “after” bag-wash sample was collected in the same way as the “before” sample but in a separate bag. Notes were made as to whether the children used soap or not when washing their hands prior to the “after” bag-wash sample being collected. During this study, additional resources such as soap or paper towels were not provided; however, schools without this equipment were not excluded from this study and children made use of the hand wash facilities and equipment normally provided to them on a daily basis.

### 2.3. Sample Analysis

Quantitative analysis was applied to the pre-intervention (right and left) bag-wash samples to test for the viable (intact) and total bacterial cell populations using flow cytometry (FCM). The samples were analyzed separately (70 mL quantity) and were then combined (100 mL quantity adjusted) so that the combined (100 mL) results could be compared to the post-intervention samples in a like-for-like quantity (100 mL).

Flow cytometric (FCM) analysis was performed on the BD Accuri C6 with CSampler according to Singh et al., 2019 [22]. Briefly, 500 μL of bag-wash sample was stained at 37 °C for 10 min with SYBR Green I (Invitrogen, Carlsbad, CA, USA) at a concentration of 1X for the total bacterial cell counts and with SYBR Green I (Invitrogen) at a concentration of 1X co-stained with 6 μM propidium iodide (PI) (BD Biosciences, San Jose, CA, USA) for the intact bacterial cell counts. The stained samples were then measured separately at medium speed using an FL1-H acquisition threshold of 800. The Eawag bacterial cell analysis template was used to detect the bacterial cell populations by electronic gating, and manual compensation was used where necessary to separate positive signals from debris, as indicated in Figure 1 [21]. The flow cytometry targeted all the bacteria present on the children’s hands and was able to indicate more than one group of bacteria. It compensated for the viable but non-culturable bacteria that would have skewed any culture-based data.

Based on previous FCM studies, the viability status of the bacteria was inferred by nucleic acid staining, which separated the bacteria into live and dead cells, with live cells further characterized as having high or low nucleic acid content [22,25,26]. The HNA-stained cells were classified as being active (alive and culturable) bacteria in a community, whilst the LNA-stained cells were classified as being metabolically active bacterial cells (viable but non-culturable) that could become dormant in adverse environmental conditions. Singh et al. (2019) reported that high levels of LNA bacterial cells were considered to be transient flora on human hands [22].

The post-intervention data per set of 100 mL bag-wash samples were analyzed for turbidity and viable (intact) and total bacterial cell populations.

The turbidity was measured in nephelometric turbidity units (NTUs). The turbidity of the hand wash samples was determined using the TN-100 Turbidimeter (Eutech, Singapore). Before the sample readings, the turbidimeter instrument was calibrated as recommended by the manufacturer and the South African Water Quality guideline standards. The primary standards included 0.02, 20.0, 100, and 800 nephelometric turbidity units (NTUs) provided along with the meter. The sample vessel was rinsed three times with at least 10 mL of the sample, after which the sample was added to the vessel up to the mark indicated on the vessel. The reading was read with the turbidimeter and used as read. Any differences in the turbidity pre- and post-handwashing was recorded along with the log10 reductions in both the live (intact) cells and total bacterial count. The turbidity counts were included to determine if it would be an easy method to estimate bacterial contamination in the hand wash samples without the need for FCM or culture-based methods, and to determine if the bacterial loads were potentially linked to dirt on the hands. In a review article published by De Roos et al. (2017), they reported that there were positive associations between drinking-water turbidity and acute gastrointestinal illness in various cities [27].

### 2.4. Data Analysis

Data were captured on an Excel spreadsheet and uploaded to IBM SPSS Statistics version 26 for statistical analysis. All the FCM data were processed with the BD Accuri C6 software version 1.0.264.21 (BD Biosciences), and boxplots were computed with GraphPad Prism 10 software. Pre-intervention, a non-parametric test was conducted (Mann–Whitney U test) to determine the differences between the results of the right and left hands and Spearman’s rho non-parametric correlation coefficient was used to assess the relationship between the total bacterial count and the intact live cells count on both the right and left hands. The same tests were conducted on the results for gender differences. Post-intervention, a Spearman’s correlation test was run to determine the relationship between the total cell count and the intact live cells before and after children had washed their hands with or without using soap. A dataset was created for the results of all the children whose hands were sampled prior to the intervention and post-intervention (N = 148) in both the IG and CG so that the pre-intervention results could be accurately compared to the post-intervention results. A Friedman test was conducted in order to measure the three points in time to obtain a mean rank and a Wilcoxon signed-rank test was used to indicate significant differences between these.

### 2.5. Intervention

Figure 2 provides a schematic representation of the intervention that was conducted on the IG within a month of the pre-intervention data collection. The intervention was administered to the children in the IG, their parents and the preschool by the researcher, who, as a qualified environmental health practitioner, was equipped to provide the intervention. An interactive learning exercise was used for the children. This was conducted by the researcher using “simulation education” with the application of GloGerm^®^ to the children’s hands to simulate germ distribution. The children rubbed GloGerm^®^ onto their hands and then placed them into a “Magic Box”, which was a black box fitted with an ultraviolet black light and a viewing hole to allow the children to see the “germs” glowing on their hands. They were then taught to wash their hands using the WHO method, and once again, placed their hands inside the box. Correct handwashing would remove all the “germs” and they would not be visible under the ultraviolet black light. The class was also given a three-week hand wash reminder chart, where the children placed stickers on the chart for every day they washed their hands correctly, and hand wash poster reminders to wash their hands at critical times. The reminder chart was used to reiterate the handwashing actions consistently for a period of time as children learn best through repetition and the motivation and reward that the stickers provide, both of which have been known to affect and assist behavior change [28].

The results regarding additional intervention aspects such as the monthly inspections and HH information relayed to parents have been previously reported [23,29] and are shown in Figure 2 to provide context.

## 3. Results

The total and intact (live) bacterial populations for the left and right hands as well as the distribution of low nucleic acid (LNA) and high nucleic acid (HNA) content bacteria are illustrated in Figure 3. The total bacterial population includes both live and damaged cells, while the intact population only represents live bacterial cells.

In Table 1, the pre-intervention bag-wash results, with the combined totals and genders, show that there was no significant difference between the right and left hands (right hand *p* = 0.167 and left hand *p* = 0.325) and no difference between the right and left hands in relation to gender (right hand *p* = 0.434 and left hand *p* = 0.296). The Spearman’s rho non-parametric correlation coefficient indicates a moderately strong correlation between the intact live cells and the total bacterial count for both the right (r_2_ = 0.832; *p* = 0.000; N = 160), left (r_2_ = 0.815; *p* = 0.000; N = 160) and combined hands (r_2_ = 0.838; *p* = 0.000; N = 160).

Post-intervention samples were drawn from 148 children (IG n = 77 and CG n = 71). A Spearman’s correlation test was run to determine the relationship between the total cell count and the intact live cells before and after the children had washed their hands with or without using soap. There is a moderate positive correlation between the total cell count and the intact live cells (r_2_ = 0.332) before the children went to wash their hands, with a similar moderate correlation (r_2_ = 0.557) in the samples taken after handwashing. All the children used soap to wash their hands if it was available. Twelve of the children from two preschools in the IG did not have soap available and so were not able to use soap to wash their hands, whereas the remaining sixty-five children from seven preschools used the soap available to them. In the CG, 42 children from five preschools did not have access to soap and 29 children from the remaining five preschools did have access to and used soap.

There was a significant correlation in the IG between the intact live cells and the total cell count before handwashing (r_2_ = 0.306, n = 77, *p* = 0.007), while after handwashing, the correlation was slightly stronger (r_2_ = 0.566, n = 77, *p* = 0.000). The CG had a similar result with a moderate correlation (r_2_ = 371, n = 71, *p* = 0.001) before handwashing and a slightly stronger correlation afterwards (r_2_ = 521, n = 71, *p* = 0.000). The mean intact live cell count in all the samples before handwashing was 12 × 10^7^, and after handwashing, it reduced to a mean of 7 × 10^6^. The test was repeated on the children who used soap (n = 99) and those who did not use soap (n = 49) regardless of which group they were in. There was a significant correlation between the live intact cells and the total cell count (r_2_ = 0.327, n = 49, *p* = 0.022) in the hand samples of the “no soap” group before handwashing, with a similar correlation (r_2_ = 0.563, n = 49, *p* = 0.000) after handwashing. The “soap” group had a moderate correlation before handwashing (r_2_ = 0.366, n = 99, *p* = 0.000), with little significant correlation afterwards (r_2_ = 0.559, n = 99, *p* = 0.000).

The turbidity before handwashing was measured in both the IG and CG, with the mean turbidity in the IG measuring 78.2179 NTU (nephelometric turbidity unit) and in the CG measuring 141.0746 NTU. After handwashing, the mean turbidity was 27.4001 NTU for the IG and 44.3429 for the CG. A decreased mean of 50.8178 NTU for the IG and 96.7317 NTU for the CG was recorded. The difference in turbidity before and after handwashing was measured in both the “soap” and the “no soap” groups. The mean turbidity in the “no soap” group before handwashing was 80.83734694 NTU and in the “soap” group was 69.33575758 NTU. The turbidity before and after handwashing in both the “soap” and “no soap” groups was analyzed. There was a mean reduction of 67.0841 NTU in the “soap” group and 80.8373 NTU in the “no soap” group. There was a significant positive correlation between the turbidity pre- and post-handwashing in the “no soap” group (*r* = 0.489; N = 49; sig. = 0.000) and the “soap” group (*r* = 0.580; N = 98; sig. = 0.000).

A dataset was created for the results of all the children whose hands were sampled prior to the intervention and post-intervention (N = 148) in both the IG and CG so that the pre-intervention results could be accurately compared to the post-intervention results. The data for the total cells and total intact live cells pre-intervention were compared with the total cells and total intact live cells post-intervention prior to handwashing as well as to the total cells and total live cells after handwashing. Table 2 indicates the results for the male and female children in the IG and CG pre-intervention, post-intervention before washing hands and post-intervention after washing hands with or without soap.

A Friedman test was conducted in order to measure the three points in time to obtain a mean rank. The results of the Wilcoxon signed-rank test indicated significant differences (*p* = 0.000) between these three results in the IG and CG for the total cell count, where the post-intervention pre-handwashing total cell count was significantly different to the pre- and final post-intervention samples in both groups (Figure 4). The same analysis was conducted on the total intact cells for the same three points in time. The result indicates a significant difference (*p* = 0.000) in the IG between the pre-intervention, post-intervention, pre-handwashing and post-intervention post-handwashing intact live cell counts. In the CG, there was no significance placed on the difference in the counts pre- and post-intervention (*p* = 0.231), but there was a significant difference (*p* = 0.000) between the post-intervention pre-handwashing and post-intervention post-handwashing intact live cell counts.

The equation below was used to calculate the percentage reduction:$$Percentage\;reduction=\frac{\left(A-B\right)\times 100}{A}$$
where *A* is the number of viable microorganisms before the treatment and *B* is the number of viable microorganisms after the treatment.

The effect size between the pre- and post-intervention intact live cell counts through the application of Cohen’s *d* indicated a medium effect (*d* = 0.42), with a small effect (*d* = 0.25) between the post-intervention pre-handwashing and post-intervention post-handwashing intact live cell counts.

## 4. Discussion

The results of the hand swab samples were compared to determine whether the microbiological counts on the hands had decreased post-intervention. The pre-intervention hand samples for both the IG and CG showed no significant difference in the cleanliness of boys’ and girls’ hands, which contradicts previous studies. These studies reported that a higher standard of hand hygiene was found in girls, especially in the under-five age group [8], and that according to a meta-analysis of health behaviors in adolescents, girls washed their hands consistently more than boys [30]. Pre-intervention hand swab samples were taken from both the right and left hands to determine whether the dominant hand carried a higher microbiological load as a result of the dominant hand being used more frequently to pick up objects [7]. According to the analysis, there was no significant difference between the dominant and other hand.

Due to the results of the pre-intervention samples, the post-intervention samples were collected by washing both hands together in a single bag. The purpose of these samples was to determine if the intervention had helped the IG to wash their hands more effectively, thereby reducing the microbiological counts on the hands in this group. In both the IG and CG, some preschools did not have soap available but all had running water and wash basins for handwashing. In schools where soap was available, the children used soap to wash their hands without prompting from caregivers or the researcher. The total cell count, active live cells and turbidity were measured in the samples from both groups, prior to handwashing and post-handwashing, keeping note of the “soap” and no soap” groups. There was a reduction in turbidity in both the IG and CG, with a positive correlation between the “soap” and “no soap” groups both pre- and post-handwashing. Higher fecal bacterial counts have been associated with higher levels of turbidity in hand bag-wash samples in previous studies [31], indicating that the reduced turbidity found in both groups regardless of whether they used soap or not showed that the simple mechanism of rubbing hands together with clean water as per the WHO’s recommended method could potentially improve hand hygiene. As there was a correlation between the total cell counts and the live active cells, in both groups, the motion of washing and rinsing the hands in clean water could remove some micro-organisms from the hands, providing a similar result to a study in Indonesia, which reported that children who washed their hands using more motions displayed a reduction in *E. coli* on the hands [32]. It has been reported that there are associations between soil microbial contamination and diarrheal diseases in children [33], making the understanding of dirty hands important, especially in children. The results of this study showed that a simple hand rinse (washing without soap) reduced the bacterial loads in the control schools comparably to the intervention schools, suggesting that the mere removal of dirt could reduce the risk of disease.

Comparative analysis of the results from the bag-wash samples of those children who were present for both the pre- and post-intervention sampling provided an opportunity to compare a robust, valid sample, as only those children who were present at both opportunities were included in the sample for this analysis. There was a significant difference in the results of the IG live intact cell counts pre- and post-intervention, which was not observed in the CG. As only the IG was exposed to the intervention, this could indicate that the intervention assisted with improved handwashing techniques, thereby dislodging live intact cells, as not seen in the CG. The method of reporting the differences in the total cell counts and intact cell counts is a novel one, as the previous literature does not report on such cell counts but rather results based on microbial growth pre- and post-intervention, as reported in a systematic review of hand contamination studies [19].

## 5. Conclusions

The purpose of this study was to discover whether a simple intervention would reduce the microbial counts on the hands of preschoolers. The total cell count and live intact cell count have been established for these preschool children based on gender, both prior to and after washing their hands. The intervention was intended to teach children the correct way to wash their hands (WHO method), making use of an entertaining and cost-effective intervention. Although there was improvement in both the intervention and control groups pre- and post-intervention, there was a significant reduction in the live cell counts in the IG, indicating that this group used the WHO method to better effect than the CG. Washing hands, either with or without soap, while using the mechanics of rubbing the hands under running water can reduce the microbiological counts on the hands of preschoolers. Therefore, reinforcing the washing of hands at critical times and ensuring correct handwashing procedures can assist in reducing hand-hygiene-related diseases in preschool children. Including handwashing with soap teaching activities for preschool children will increase compliance and ensure that no transient microbes, which could cause disease, remain on the hands.

Several limitations were identified during this study, notably the lack of consistency in the activities of the children from one school to another during the bag-wash sample collection process. Each school was a private facility, following their own programs and curriculums. During sample collection, some children were drawing or writing; in other classes, they may have been painting; and in yet others, it was story time, which could have influenced the results of the samples. In addition, the left- and right-handedness of the children was not noted during the pre-intervention sampling, and although the results showed no significant differences between the two, there are other aspects such as cultural behaviors of using the dominant versus non-dominant hand for certain activities. As the sample population was drawn from a cohort of regulated and compliant preschools, it provided an opportunity to study hand hygiene compliance in a well-resourced setting. There is an opportunity for a future study to provide a comparison between suburban, well-resourced preschools and those in rural and underserved communities to determine whether an intervention such as this could positively influence the microbiological counts.

## Figures and Tables

**Figure 1 ijerph-22-00846-f001:**
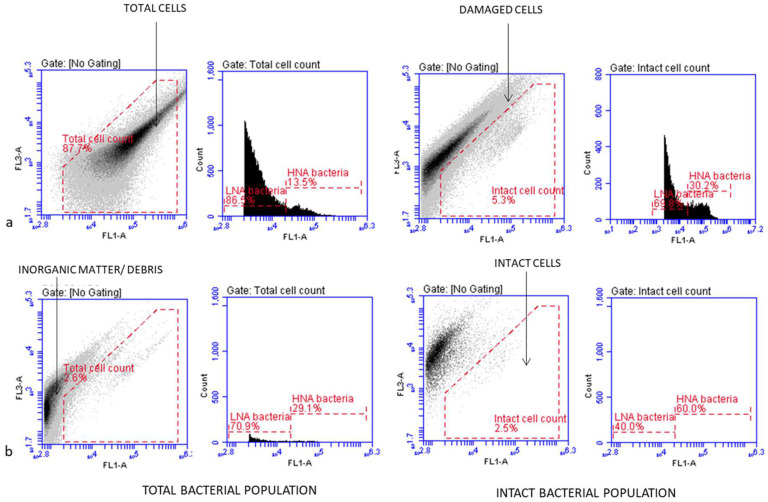
Flow cytometric dot plots and histogram examples of the distribution of bacterial cells in a hand wash sample, showing the data collection and the gating strategy as described by Singh et al. (2019) [22]. The bacterial population in the hand wash sample is shown (**a**) before the intervention and (**b**) after the intervention.

**Figure 2 ijerph-22-00846-f002:**
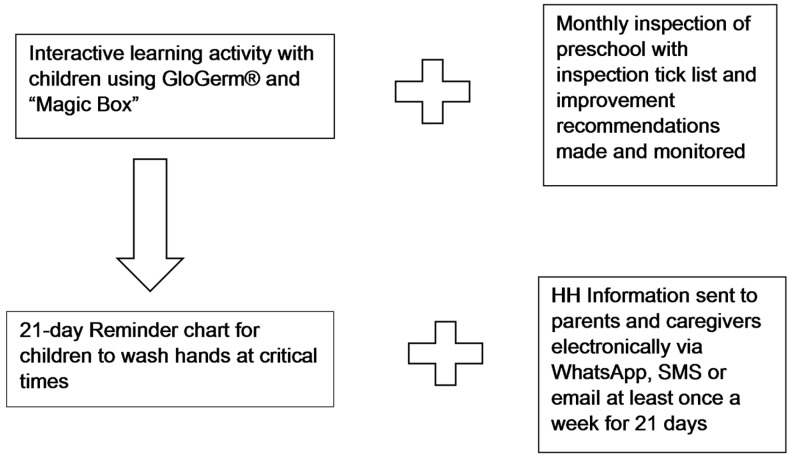
Schematic representation of the hand hygiene intervention.

**Figure 3 ijerph-22-00846-f003:**
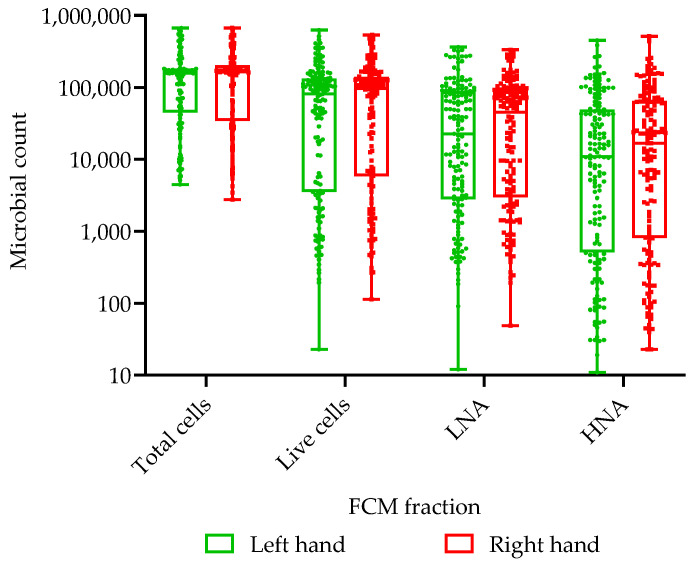
**Whisker-box plot comparing the** pre-intervention total bacterial and live (intact) microbial cell counts for the right and left hands.

**Figure 4 ijerph-22-00846-f004:**
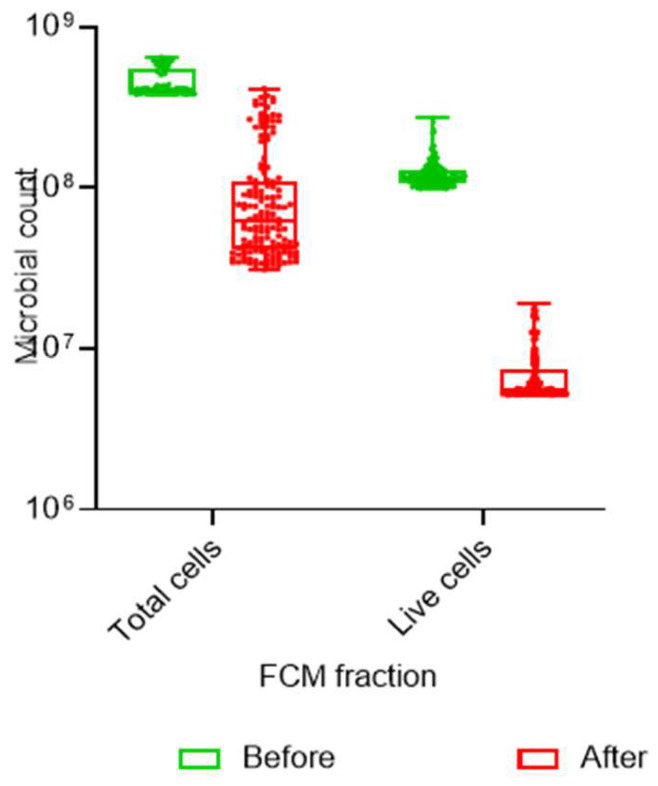
**Whisker-box plot** of the results obtained post-intervention by flow cytometry.

**Table 1 ijerph-22-00846-t001:** Pre-intervention total bacterial and intact live cell count for both hands per gender.

Gender	No.	Total Cell Count/70 mL (RH)	Intact Live Cells/70 mL (RH)	Total Cell Count/70 mL (LH)	Intact Live Cells/70 mL (LH)
Male	81 (51.0%)	2.14 × 10^6^	1.39 × 10^6^	1.99 × 10^6^	1.25 × 10^6^
Female	79 (49.0%)	2.63 × 10^6^	1.59 × 10^6^	2.46 × 10^6^	1.51 × 10^6^

**Table 2 ijerph-22-00846-t002:** Pre- and post-intervention bag-wash results.

	Pre-Intervention	Pre-Intervention Male	Pre-Intervention Female	Post-Intervention	Post-Intervention Male	Post-Intervention Female	Pre-Intervention Mean	Post-Intervention Pre-HW Mean	Post-Intervention Post-HW Mean	Pre-HW with Soap	Post-HW with Soap	Pre-HW No Soap	Post-HW No Soap
No. of samples	160	81 (51.0%)	79 (49.0%)	148	71 (48.0%)	77 (52.0%)	148	148	148	98	98	49	49
Total bacterial count IG CFU/mL		3.17 × 10^6^	3.68 × 10^6^		4.9 × 10^7^	4.65 × 10^6^	3 × 10^8^	4 × 10^8^	1 × 10^8^	4.6 × 10^8^	1.1 × 10^8^	6.0 × 10^8^	1.3 × 10^8^
Total bacterial count CG CFU/mL		2.74 × 10^6^	3.29 × 10^6^		4.53 × 10^6^	4.35 × 10^6^	3 × 10^8^	4 × 10^8^	8 × 10^7^	4.0 × 10^8^	7.0 x 10^7^	4.6 × 10^8^	9.2 × 10^7^
Intact live cells CFU/mL							2 × 10^8^	1.2 × 10^8^	7.0 × 10^6^	1.2 × 10^8^	6.8 × 10^6^	1.2 × 10^8^	7.4 × 10^6^
Intact live cells IG CFU/mL		2.09 × 10^6^	2.48 × 10^6^		1.26 × 10^6^	1.26 × 10^6^	2 × 10^8^	1 × 10^8^	7 × 10^6^	1.2 × 10^8^	7.0 × 10^6^	1.2 × 10^8^	1.0 × 10^7^
Intact live cells CG CFU/mL		1.57 × 10^6^	1.94 × 10^6^		1.24 × 10^6^	1.17 × 10^6^	1 × 10^8^	1 × 10^8^	6 × 10^6^	1.2 × 10^8^	6.0 × 10^6^	1.1 × 10^8^	7.2 × 10^6^
Turbidity NTU								108.3	35.4 (*p* < 0.001)	102	33 (*p* < 0.001)	121.1	40.3 (*p* < 0.001)
Turbidity NTU IG								78.2	27.4 (*p* < 0.001)	76.4	26.6 (*p* < 0.001)	96	35.2 (*p* < 0.001)
Turbidity NTU CG								138.9	44.3 (*p* < 0.001)	163.8	49.1 (*p* < 0.001)	125.3	41.1 (*p* < 0.001)

## Data Availability

Data are available on request from the authors.

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
