# Peer review of "Monitoring the Effectiveness of a Preschool Hand Hygiene Intervention Using Flow Cytometry to Determine the Reduction in the Microbiological Counts"

_ijerph, 2025, doi:10.3390/ijerph22060846_

Round 1
Reviewer 1 Report
Comments and Suggestions for Authors
Overall this paper was well written and the study was of good quality. Although hand hygiene is a simple tool used to prevent spread of organisms and diseases, many people still do not perform appropriate hand washing.
Your study was very informative and suggest that the quality of hand hygiene in a vulnerable population is very important. It was very good to include children who had no soap to wash hands and that the process of washing hands with water and friction removed microbes from hands.
The testing method was explained very well with supplemental material to help the reader understand the scientific method for calculating or measuring hand samples to count microbes.
Comments on the Quality of English LanguageOverall the quality of the English language was very good in this article. In the introduction, there were a few areas with grammar, although they were not incorrect but can be adjusted to make the reading easier to understand.
Line 34 remove the words "of the", it easier to say Two leading causes of death
Line 35 used the word gastric and respiratory infections since you are talking about two body systems.
Line 44 misspelled the word thought, missing the letter "t".
Line 45 found in girls under five years of age, "remove especially in the"
Line 50-53, very long sentence consider making two shorter ones.
Study design and sample collection was good.
Good sample and data analyses.
Excellent intervention and the explanation of why your team chose this type intervention for the children.
Results were good and thanks for the supplemental material which helped me understand the technique used to measure the microbes from the children's hands.
The discussion was very well done and explained the results of the study very well.
Line 269 the letter "p" is missing from pre and post-test.
Author Response
Author’s reply:
Thank you for your comments and corrections. These have been addressed in the manuscript.
Line 34 remove the words "of the", it easier to say Two leading causes of death Corrected
Line 35 used the word gastric and respiratory infections since you are talking about two body systems. Corrected
Line 44 misspelled the word thought, missing the letter "t". Corrected
Line 45 found in girls under five years of age, "remove especially in the" Corrected
Line 50-53, very long sentence consider making two shorter ones. Corrected
Line 269 the letter "p" is missing from pre and post-test. Corrected

Reviewer 2 Report
Comments and Suggestions for Authors
I thank the editor and authors for the chance to review this paper.
This is a useful study, in particular with it being based in South Africa. The sample number of children participating is large enough to make conclusions. There is a rigour to the way the data is presented and analysed. The conclusion makes it clear that, in this case, the intervention aided the reduction of bacteria.
Whilst the paper doesn’t make a radically new contribution to knowledge it offers incrementally useful knowledge to the field.
My main areas to focus on for a revision are as follows:
Gap in knowledge
Consider why wouldn’t a hand hygiene intervention reduce microbiological count? It would be useful to add a short paragraph comparing this study with others in the introduction– where is the knowledge gap that this paper fills? Is it related to South African studies or other studies that use different interventions?
Intervention
More clarity is required regarding the intervention. The intervention consists of 3 components – e.g. Glogerm demonstration, Handwashing demonstration and a 3-week chart. Why was the intervention designed this way and do the researchers have a sense of which element was most effective or whether all three components are required to see results. Figure 2 contains some elements that are not described in the section above so it’s would clearer to have this outlined. It’s also not clear when the intervention took place – e.g. immediately after the pre-intervention sample collection? Adding this reflection/detail will help other researchers and intervention designers.
Soap
Line 68
Which resource availability was considered? More detail would be useful here? E.g capacity to do the training? Availability of water and soap? How did the researchers ascertain that the children used soap and how reliable was this? (Line 189).
Sample Collection
According to the sample collection section a pre-intervention study involved taking samples of bacteria from the children immediately from their current activity – this then would capture their ‘natural’ state that could contain a large build up of bacteria.
The Post Invention sample took place 2 months later but samples were taken immediately on their arrival school (unlike the pre-intervention study). Why was this the case? Is this related to seeing the effect of parent education on home hand-washing? Wouldn’t there be more parity is the sample method was identical (same time of day/class) to the pre-intervention sample? In ‘after’ if children were then asked to wash their hands, isn’t it inevitable that there would be a difference between both ‘before’ and ‘after’ as well as ‘pre’ and ‘post’? This section needs more clarity to ensure rigour of experimentation.
It's also not clear to the reader how ‘before’ and ‘after’ post-intervention measures (2 measures) make one ‘post intervention’ results.
Conclusion
It would be useful to know the limitations of the study and further research questions.
Written English
Broadly the work is well written. Minor corrections needed – e.g. line 72 missing ‘to’ before ‘participate’
Author Response
Thank you for your comments and questions. The intervention and sample collection sections have been addressed to ensure a better understanding for the reader.
Gap in knowledge
Paragraph inserted as suggested Line 70-77
Intervention
Line 192
See Lines 201-210
Soap
See Lines 122-124 and 192-194
Sample Collection
The contradicting statements in this section have been corrected in Lines 102-124
Conclusion
The conclusion has been strengthened, and limitations have been included in Lines 306-325

Reviewer 3 Report
Comments and Suggestions for Authors
The finding of this study is predictable i.e. that hand washing reduces the microbiological count on hands. However, this would apply regardless of age. The main issue in young children is compliance and, as stated, placing objects in the mouth.
Appropriate consent was obtained. The authors should state if the sampling was done at the same time of day when presumably the children were engaged in similar activities to other days or whether the time varied. Flow cytometry was used rather than culture so no bacterial species or relative numbers of each species were available but included non-culturable bacteria. The authors should explain what ‘groups’ of bacteria were identified – is this the low and high nucleic acid content bacteria mentioned in results? The terms should be explained particularly which species would be expected in each group. Was the children’s use of soap random or only related to the availability of soap? Was lack of soap a marker of the schools’ overall compliance with the study? GloGerm is commonly used in hand hygiene training but reduction in compliance occurs within a few weeks of training unless repeatedly reinforced. In this study the reminder was at 21 days and a monthly inspection. Most pathogens would be expected in transient rather than resident flora. The sampling would not distinguish these categories but the easier to remove would be the transient flora. Turbidity would be affected by the activity the children were performing prior to sampling. What was the proportion of right and left handedness?
The finding that teaching hand hygiene improves cleanliness of hands is not surprising. However, more information is needed on the effects of parental teaching, preceding activity and any correlation between parental interest and availability of soap.
The abstract contains several unnecessary hyphens. ‘Faecal’ is misspelt on p2.
Author Response
Thank you for the questions and comments, and these have been corrected in the manuscript.
The authors should state if the sampling was done at the same time of day when presumably the children were engaged in similar activities to other days or whether the time varied. This has been clarified in Lines 102-111
Flow cytometry was used rather than culture so no bacterial species or relative numbers of each species were available but included non-culturable bacteria. The authors should explain what ‘groups’ of bacteria were identified – is this the low and high nucleic acid content bacteria mentioned in results? The terms should be explained particularly which species would be expected in each group
The reviewer is correct that simple handwash actions would reduce the microbial load, but the study approach was to determine if the reduction was due to removal of the bacteria, inactivation of the bacteria or death of the bacteria. This has now been included in the manuscript as well as a more detailed explanation of the bacterial groups and an explanation of the use of turbidity. Lines 148-155 and Lines 168-173
Was the children’s use of soap random or only related to the availability of soap? Was lack of soap a marker of the schools’ overall compliance with the study? An additional sentence has been included in Lines 122-125 and Lines 266-267.
Most pathogens would be expected in transient rather than resident flora. The sampling would not distinguish these categories but the easier to remove would be the transient flora. Turbidity would be affected by the activity the children were performing prior to sampling. This has been explained in Lines 280-285
What was the proportion of right and left handedness? This was not documented but it has been included in the limitations in Lines 319-322
The finding that teaching hand hygiene improves cleanliness of hands is not surprising. However, more information is needed on the effects of parental teaching, preceding activity and any correlation between parental interest and availability of soap.
We do agree that a routine teaching of hand hygiene will increase compliance and that access to soap would ensure that no transient microbes that cause disease would remain on the hands. The parental involvement has been addressed in a previous publication: S. Lange, T.G. Barnard and N. Naicker, “The effect of a hand hygiene intervention on the behaviour, practices and health of parents of preschool children in South Africa,” Perspectives in Public Health. 2022 Nov;142(6):338-46, doi: 10.1177/17579139221123404.
The abstract contains several unnecessary hyphens. ‘Faecal’ is misspelt on p2. Corrected

Round 2
Reviewer 3 Report
Comments and Suggestions for Authors
The comments have been addressed and limitations explained.